# Vitamin D Fortification and Its Effect on Athletes’ Physical Improvement: A Mini Review

**DOI:** 10.3390/foods12020256

**Published:** 2023-01-05

**Authors:** Cong Feng, Xinjie Song, Meram Chalamaiah, Xiaofeng Ren, Mingxing Wang, Baoguo Xu

**Affiliations:** 1Department of Physical Education, Jiangsu University, Zhenjiang 212013, China; 2School of Biological and Chemical Engineering, Zhejiang University of Science and Technology, Hangzhou 310023, China; 3Department of Agricultural, Food and Nutritional Science (AFNS), 4-10 Ag/For Centre, University of Alberta, Edmonton, AB T6G 2P5, Canada; 4School of Food and Biological Engineering, Jiangsu University, Zhenjiang 212013, China; 5Department of Business Administration, School of Business, Yeungnam University, Gyeongsan 38541, Republic of Korea

**Keywords:** vitamin D, nutritional supplement, athletes, human health, biofortification

## Abstract

Poor vitamin D status is a widespread problem regardless of age and sex, emphasizing the necessity of new food sources to improve vitamin D levels. Currently, approximately 60% of dietary vitamin D consumption occurs via fortified foods. Vitamin D insufficiency (50–90%) is widespread according to age and region, despite different levels of sunlight exposure. The food industry must identify more effective strategies to increase normal dietary vitamin D intake and improve overall health. Strategies for vitamin D fortification include bioaddition, wherein a vitamin D-rich food source is added to staple foods during processes. These bioadditive strategies expand the range of vitamin D-containing foods and appeal to different preferences, cultures, and economic statuses. In several countries, vitamin D deficiency places athletes at a high risk of disease susceptibility. Due to low sun exposure, athletes in countries with higher and lower levels of sunlight have similar risks of vitamin D deficiency. In this review, we summarize recent technical advances to promote vitamin D utilization by humans during sports activities and in relation to the normal practices of athletes.

## 1. Introduction

Nutrition is a well-recognized and invaluable component of athletic training and all competitive programs [1]. A sufficient and well-balanced nutrient supply helps athletes achieve optimal performances. Compared with nonathletes, athletes have higher levels of physical activity and greater energy requirements [2,3,4]. For all physically active people, nourishing supplement is the most significant complementary component to the physical activity. Sufficient nutrition complements both training and recuperation and can simultaneously enhance metabolic adaptation to training. Generally, requisite energy can be derived from different types of macronutrients (carbohydrate, protein, and fat) and micronutrient supplements. Achieving a balanced energy level is important for athletes with high energy demands due to intense physical activity, and dietary sports supplementation contributes to performance, adaptations, and recovery from training [5]. Vitamins, minerals, and micronutrients are acknowledged as essential nutrients that provide health benefits and ergogenic action in both normal and physically active individuals.

Micronutrient deficiency is a crucial public health problem which can cause compromise immune systems, interfere with child growth, and affect human potential worldwide [6]. This type of malnutrition can lead to various infectious or chronic diseases and, thus, affect quality of life and epidemiological parameters such as morbidity and mortality [7]. Consequently, micronutrient deficiency-related malnutrition highly affects biochemical and physiological functions and is related to premature death, disability, and a reduced work capacity [8]. Athletes may experience decreased performance and compromised health if they do not receive adequate nutrition after a high volume of training [9,10].

In recent years, both developing and developed countries have faced an increased incidence of several human diseases related to vitamin D deficiency and/or unavailability in the daily diet. Vitamin D deficiency is a global phenomenon that affects both athletes and nonathletes [11]. Athletes have been found to harbor suboptimal vitamin D levels, regardless of the national level of sunlight exposure, and this deficiency has been associated with etiological factors such as ultraviolet B sunlight availability and exposure, skin color, dietary practices, and topical sunscreen use [12]. According to recent literature, 25-hydroxyvitamin D [25(OH)D] deficiency is associated with reduced signaling of pathways necessary for growth and survival, including those involving mitogen-activated protein kinases, Src, and Akt or protein kinase B, and impaired muscle cell development [13]. Vitamin D may be protective against cardiovascular disease, cancer, and other diseases, although additional evidence is needed to support these claims [14]. Vitamin D supplementation is reported to facilitate increased muscle strength and mass. Moreover, vitamin D supplementation can help maintain an appropriate balance in the levels of steroid hormones, including testosterone, in the serum [15].

Previous studies on athletes have indicated a direct relationship between vitamin D levels and performance parameters, including velocity, jump height, muscle tone and force, and handgrip strength [16,17,18]. Similarly, a few studies have shown that combined supplementation with vitamin D and calcium reduces the rate of stress fracture [19]. The requirements for vitamin D and other essential micronutrients, including antioxidants, suggest that the absorption of these nutrients plays an important role in athletes’ development of physical capacity. Industrial methods of producing low-calorie and fat-free dairy products remove fat, including vitamin D, from whole milk.

Food fortification has gained significant interest as a nutraceutical strategy for addressing micronutrient malnutrition. This strategy has the binary advantage of enabling the distribution of nutrients to large segments of the population without radical changes in food consumption patterns. Compared with other interventions, food fortification is likely to be more cost effective; moreover, the regular consumption of fortified foods advantageously enables the maintenance of consistent physiological body stores of certain targeted micronutrients [20,21]. In this review article, we discuss the requirements, metabolism, importance, and understanding of fortification with vitamin D in the context of sports nutrition and the possible roles of these micronutrients in improving athlete performance.

## 2. Nutritional Supplements for Athletes

### 2.1. Essential Dietary Supplements

Providing educational information about nutrition to athletes and coaches as well as diet targeting optimal performance and recovery are very important for sports dietitians and nutritionists. Currently, dietary supplement use is widespread among the athletic population, although the overall requirements for the efficacy of certain important ingredients remain controversial. Dietary supplements can play a crucial role in ensuring that athletes consume appropriate amounts of calories, macronutrients, and micronutrients [22]. These essential dietary supplements do not alter health nutrition. Studies of numerous dietary components have explored the potentially important effects on various aspects of athletes’ health, including improvements in training, performance, and recovery [23]. Sports nutritionists and dietitians must have a detailed understanding of nutrition, exercise, and performance that would enable athletes to understand the positive and negative results of various studies related to dietary supplements [24]. All sports nutrition professionals must be able to assess the scientific value of studies and advertisements about exercise and nutritional products. In the literature, investigations primarily focus on whether the proposed supplement has been found to influence training and exercise adaptation by enhancing muscle hypertrophy and subsequent ergogenic potential. A few ingredients may exhibit a limited potential to trigger training adaptation or exert an ergogenic effect; however, these ingredients may affect muscle recuperation or provide health advantages that may be useful in certain populations [24].

### 2.2. Importance of Vitamin D for Active Sports

Vitamin D deficiency is common among both athletes and nonathletes. This vitamin has been reported to play pivotal roles in health and performance [25,26]. An updated systematic review and meta-analysis reported a prevalence of vitamin D inadequacy of 56–95% among athletes worldwide [27]. Several studies conducted in Middle Eastern countries reported a high prevalence (84%) of vitamin D deficiency among athletes and inadequate serum 25(OH)D concentration of <75 nmol/L [28]. A recent study performed in Tunisia reported an elevated prevalence (>90%) of vitamin D deficiency and inadequate serum concentrations (<75 nmol/L) among healthy young athletes during the winter season, as well as significantly lower vitamin D levels in indoor athletes than in outdoor athletes [28].

Athletes should have higher serum vitamin D concentrations than nonathletes since they tend to engage in outdoor activities. Overall, an increased incidence of injury has been associated with a lower serum 25(OH)D concentration among athletes [29]. Moreover, a significant positive association was identified between vitamin D intake via dietary sources and/or prescribed vitamin D supplements and the serum 25(OH)D concentration. Experts in sports therapy play a significant role in expanding athletes’ awareness about the significance of vitamin D for performance and health (e.g., preventing damage). Figure 1 depicts the main mechanism of vitamin D synthesis in the human body; it is synthesized from substrate 7-dehydrocholesterol in the skin under the action of ultraviolet light, and then affects the physiological functions of the human body. Alkoot et al. (2019) [28] observed a need to increase awareness about the extreme incidence of low vitamin D intake among athletes in Kuwait and elsewhere in the Middle East, specifically during Ramadan. These authors also proposed a requirement for annual medical assessments based on the following experimental findings: (i) implementation, (ii) athletes in developing countries face a high consequence of vitamin D inadequacy, (iii) the prevalence of vitamin D deficiency is high during extreme winters and summers, and (iv) prolonged exposure to the sun diminishes the protective benefits of vitamin D against hypovitaminosis D and other age-associated disorders. A recent study by Harju et al. reported that greater attention to prevention and treatment is needed because vitamin D insufficiency is high in elite athletes [30].

The Institute of Medicine has officially established vitamin D adequacy threshold levels for the North American population, given the relationship between vitamin D status and bone health maintenance [30,31,32]. In Western countries, including Canada and the United States, comparisons with white and nonwhite respondents in studies such as the Canadian Health Measures Survey have yielded similar results [3]. A high prevalence of vitamin D deficient nutrition status has been reported worldwide. For instance, the prevalence of vitamin D deficiency has been reported to be 24%, 37%, and 40% in the United States, Canada, and Europe, respectively [33,34]. Supplementation and food fortification are two strategies to combat vitamin D deficiency within a society, but food fortification has obtained more attention due to the increasing levels of vitamin D intake [35]. It was projected that a large amount of vitamin D intake is from fortified foods in the United States [36].

### 2.3. Clinical Complications and Solutions with Better Vitamin D Fortification

Globally, most people avoid sunlight exposure to remain cool, prevent sun-related skin aging, reduce the risk of skin cancer, and avoid undesired skin tanning. Consequently, health concerns caused by vitamin D deficiency have become a challenge worldwide [37,38] among people of all ages [39]. Recently published evidence demonstrates the relationship of hypovitaminosis D with higher morbidity and mortality. Vitamin D deficiency is more prevalent among Africans relative to Caucasian populations [40]. Webb et al. (2018) [41] observed that people from the UK with type V or brown skin required approximately 25 min of early afternoon daylight each day to achieve sufficient serum D levels, whereas those with lighter skin required shorter durations. Furthermore, Holick (2020) [42] reported that vitamin D deficiency caused by insufficient intake from food sources could lead to growth retardation and rickets in children and may be related to serious consequences including an increased risk of common cancers, autoimmune diseases, infectious diseases, and cardiovascular disease. Garland and Gorham (2017) [43] evaluated the connection between cancer and vitamin D deficiency. Their investigations emphasized that people living at higher latitudes had less exposure to sunlight, which increased the risks of both vitamin D inadequacy and cancer. Garland and Gorham (2017) [43] reported a positive association between residence at a high latitude and cancer death rates. Furthermore, meta-analyses demonstrated an increased risk of colorectal cancer in people with low serum D concentrations. Zhu et al. also reported that women with a higher 25(OH) D level had a lower risk of breast cancer than women in the middle group, and 25(OH)D was associated with an increased risk of colorectal and breast cancer but not overall cancer risk [44].

Recently, because of lifestyle factors and insufficient daylight exposure, vitamin D inadequacy has attracted more attention due to the observed relationships with the risks of severe chronic diseases. As delayed exposure to sunlight is linked with the risk of skin cancer, food fortification has emerged as a meaningful alternative to mitigate vitamin D deficiency [45,46]. Suggested approaches for improving vitamin D status have included living a healthier everyday life and weight loss to mobilize vitamin D and its metabolites from adipose tissue; however, these approaches were not helpful. Similarly, encouraging the intake of foods that naturally contain vitamin D also showed no health benefits. Good dietary sources of vitamin D, such as fatty fish, salmon and cod liver oil, are limited and not frequently consumed in the diet [46]. Consequently, food is not a main source of vitamin D, given the limited availability of natural food sources and the removal of the vitamin during processing. Currently, consumption of vitamin D supplements is not observed at the population level worldwide, and research in Greece showed that vitamin D intake is below the average requirements [10,47]. A small study from Switzerland also demonstrated that vitamin D intake was below the recommendations, and the main dietary sources of vitamin D were fish (35.2%) and dairy products (32.3%) [48]. Many national supplementation programs have cited that vitamin D fortification is the main strategy to achieving adequate vitamin D intake [49]. However, the fortification of foods with vitamin D has been suggested as an effective method, and several countries have performed either mandatory or voluntary food fortification to eliminate the disorders linked to vitamin D deficiency [50]. The foods most responsible for contributing vitamin D differ among countries according to dietary patterns and fortification policies [51].

Based on these surveys, one may conclude that all humans, including athletes, fail to absorb sufficient vitamin D and other micronutrients. Recent approaches to food fortification aimed to improve the techniques and, thus, achieve better stability and effectiveness. This review summarizes the modern interpretation of vitamin D food fortification approaches and their health impacts. Both the United States and Canada allow the voluntary fortification of some foods with vitamin D. In both countries, novel approaches to dietary vitamin D fortification include bioaddition, in which food staples are fortified by adding an additional vitamin D-rich food in the course of production or the manipulation of food preprocessing or postharvest. These bioadditive methods increase the scope of dietary supplies of vitamin D and may therefore vary by inclination, economic status, and culture. However, recent studies on the adequacy and safety of bioaddition should be conducted in various targeted populations.

## 3. Vitamin D Food Fortification: Practical Approach and Efficiency

### 3.1. Vitamin D Fortification Strategies

The fortification or enrichment of foods can be described as the increase in key nutrients and minerals to address dietary differences in the population [51,52]. Industrialized countries have a long history of food fortification to monitor for inadequacies in vitamins A, B, and D, as well as other micronutrients such as iron and iodine. In the present scenario, foods intended for adolescents are fortified with micronutrients, particularly iron, with the aim of reducing the associated health illnesses. The addition of folic acid to wheat is common in the United States, and this method has also been implemented by many nations, including Canada and Latin American countries. Food fortification can play a significant role in eradicating the development of vitamin D deficiency and malnutrition in several countries [51,52]. The General Principles of the Codex Alimentarius define food fortification as the accumulation of one or more essential nutrients in the food, irrespective of its normal status in food, to avoid or assess a proven nutrient deficiency in the specific or general population [52]. The principles also assert the quality of food during all the essential steps, including processing, storage, shipping, and delivery to consumers. Fortificants, i.e., the forms of micronutrients utilized to fortify foods, differ in cost, biological significance, and type of food. During the developmental phase, foods considered for fortification must be easy to produce, of good quality, and enjoyed by consumers. The “vehicles” for food shortages of fortification are shown in Figure 2; people’s health could be improved by ingesting food fortified with micro-nutrient elements, especially athletes’ health.

In addition, certain types of fortificants, accurately described as enrichments or micronutrients, are added to foods to replace those lost during processing [34]. Micronutrient malnutrition is a common and acute disorder in the developing world but may also signify a public health problem in developed countries [53]. Nevertheless, the combined information regarding vitamin D fortification for athletes from the literature does not fully address the additional advantageous effects. Compared with other interventions, food fortification may be more cost effective; moreover, the regular consumption of fortified foods could advantageously promote the maintenance of stable micronutrient stores in the body. Scientists analyzed the effects of single, dual, and multiple micronutrient fortification strategies on various outcomes, as shown in our concept-based framework (Figure 3). These micronutrients were administered to the targeted population through one of three food vehicles (staples, condiments, or processed foods).

Cholecalciferol, ergocalciferol, and their 25-hydroxylated metabolites are utilized in the fortification of several foods. Studies have highlighted the consequences of several intervention trials concerning the qualified efficacies of vitamins D2 and D3 in terms of increasing 25(OH)D levels and have mostly supported the latter as substantially effective for this objective [53,54]. Maurya and Aggarwal reported a vitamin D-encapsulated nanostructured lipid fabrication that could be applied in a milk yoghurt-based beverage based on a sensory evaluation [55]. Hasanvand et al. produced a fortified milk modified with vitamin D-entrapped starch-based nanoparticles, and the fortified milk had no significant difference from unfortified milk with respect to the results of the sensory analysis [56]. Golfomitsou et al. developed oil-in-water edible nanoemulsions and used them as carriers for vitamin D in fortified whole-fat milk. Fortified whole-fat milk could remain stable against particle growth and gravitational separation for at least ten days [57]. Brown et al. (2013) modeled the effects of various fortified carrier foods, such as bread, milk, and orange juice, on vitamin D levels, as adults in Germany usually have lower vitamin D levels than the corresponding population in the United States [58]. This fortification aimed to avoid deficits in the entire population while certifying that consumption by people would remain below the upper threshold values.

Grønborg et al. (2019) [59] developed a classified model of dietary vitamin D consumption by determining the contributions of vitamin D from fortified foods in a population aged 18–55 years. The results indicated that a secure intake level can be accomplished and can reach 100 μg (the EFSA upper limit for intake) only through vitamin D-fortified daily intake. Moulas and Vaiou (2018) [60] reported that traditional and biotechnological methods could be used to produce innovative and unique vitamin D-rich or vitamin D-fortified foods. The convenience of a larger range of fortified foods for daily consumption as part of a “daily vitamin D” public health policy can lead to improvements and the deterrence of vitamin D deficiency within a population. Additionally, rigorous supplemental foods are being established and launched in advanced countries to ensure that children and adolescents receive greater micronutrient supplementation. Nevertheless, micronutrient malnutrition is common and acute in the developing world; it can also signify a public health dilemma in more industrialized countries.

### 3.2. Effects of Fortified Foods on Bone Health in Athletes

Fortified dairy foods could improve bone health by reducing bone turnover. Several assessments corroborate the significance of measuring spreading factors reflective of bone remodeling as a means of evaluating the effects of fortified foods. As indicated in a recent review, clinical trials targeted at assessing the consequences of nutritional products on bone must utilize surrogate endpoints to assess the anti-fracture efficacy [61]. Fragility fracture-related quantities comprise specific hormonal factors such as parathyroid hormone (PTH) and insulin-like growth factor-I, as well as bone turnover markers related to bone formation (e.g., osteocalcin, total procollagen type 1 N-terminal propeptide, alkaline phosphatase) or bone resorption (e.g., collagen type 1 cross-linked N-telopeptide, collagen type 1 cross-linked C-telopeptide, tartrate-resistant acid phosphatase 5b) [62]. The extent of biomarker turnover within a few weeks or months following the beginning of an intervention can be used to calculate the long-term risk of fragility fracture and amount of bone loss. Vitamins D2 and D3 are translated to 25(OH)D (i.e., calcifediol) in the liver. Subsequently, 25(OH)D is changed into the highly effective metabolite 1,25-dihydroxyvitamin D [calcitriol; 1,25(OH)2D] via α-hydroxylation under a light-controlled procedure in the kidney [54]. Typically, the 1,25(OH)2D level tends to stay in the regular range, irrespective of poor vitamin D ingestion. This balance is achieved by changes in the PTH concentration, resulting in bone loss due to vitamin D deficiency [63]. In a study of well-trained professional football players, Ksiazek et al. (2017) [13] concluded that a slight correlation existed between the serum 25(OH)D concentration and muscle stiffness and elasticity when measured immediately after the transition period. In Germany, a similar result was observed in a population study by Perna [64], and a more recent report provided a novel and detailed mathematical bottom-up model of 25(OH)D concentrations [65,66,67]. This review aimed to use this knowledge to develop a novel vitamin D fortification model that would consider all vitamin D sources [65,68].

### 3.3. Emulsion-Based Nanofortification for Rapid Vitamin D Supplementation in Athletes

Emulsion-based delivery systems, such as nanoemulsions, have been suggested as ideal vectors that are especially appropriate for capturing, defending, and delivering both lipophilic and hydrophilic bioactive components in different food products [69]. Nanoemulsions are colloidal dispersions characterized as liquid droplets (general size range: 50–500 nm) in a different non-miscible incessant liquid phase. Nanoemulsions are kinetically stable throughout the storage phase and can be achieved at low surfactant concentrations; thus, they have several important applications in the food industry. Recently, vitamin D deficiency has been recognized as a global problem linked to several disorders in children and adults with considerable economic cost consequences [70]. For most humans, sunlight exposure is a major source of vitamin D. Golfomitsou et al. (2018) [57] reported the emulsion-based fortification of whole-fat milk in an attempt to boost the vitamin contents of numerous food products. Electron paramagnetic resonance spectroscopy of a stable free radical was performed to estimate the radical scavenging activities in captured vitamin D3 and fortified milk samples. The development of food-grade O/W nanoemulsions as carriers of vitamin D may represent an important strategy in the fortification of emulsion-based food matrices.

### 3.4. Biofortification

Novel strategies to supplement foods with vitamin D have incorporated the concept of bioaddition. This concept differs from biofortification, which refers to the production of food crops containing more significant levels of nutrients via genetic alteration (e.g., preharvesting) or changes in plant breeding. Ultraviolet light exposure of edible mushrooms is an example of bioaddition [65,66]. Traditional fortification approaches, in which foodstuffs are supplemented with exogenous vitamin D, will remain a significant strategy for the expansion of vitamin D consumption. However, novel fortification approaches, such as biofortification or bioaddition, have elicited considerable attention. Eggs are considered the most thoroughly examined vitamin D biofortified food. A study from Duffy et al. showed that 25(OH)D_3_-enriched fodder could significantly improve the 25(OH)D_3_ concentration in egg yolks [67]. Burild et al. (2016) [68] noticed that the vitamin D metabolite content in pork meat changes with the consumed form of vitamin D. Although several research papers have indicated a role for vitamin D in type 2 diabetes pathogenesis and treatment [69], limited data regarding the diabetes-preventing impacts are available for vitamin D-fortified foods.

## 4. Conclusions

Vitamin D, an essential nutrient, maintains blood phosphorus and calcium concentrations within a narrow physiological limit by improving the intestinal absorption of these minerals and boosting their renal reabsorption. Foods can be effectively enriched with vitamin D through the utilization of bioadditive processes that result in high vitamin D contents. Fortification also affects the nutritional health of the target population. Fortification plans alone cannot overcome various micronutrient deficiencies. Future studies should consider the efficient fortification of new products, the stabilization of vitamin D in shelf-stable products, the invention of new industrial fortification methods from the perspectives of stability and convenience, and the development of suitable packaging systems. This review summarizes recent technical advances that promote vitamin D utilization by humans during sports activities in relation to the normal practices of athletes. The data indicated that adequate vitamin D intake contributes to the health and athletic performance of athletes. However, we have limited information regarding the effect of vitamin D oversupplementation on health and performance, and more studies are needed to determine whether vitamin D oversupplementation affects the health of athletes and to explore effective ways of supplementing vitamin D-fortified foods. This review could help sports dietitians and sports physicians regularly evaluate the vitamin D status of athletes and to make appropriate recommendations to guide the athletes to achieve their best performance.

## Figures and Tables

**Figure 1 foods-12-00256-f001:**
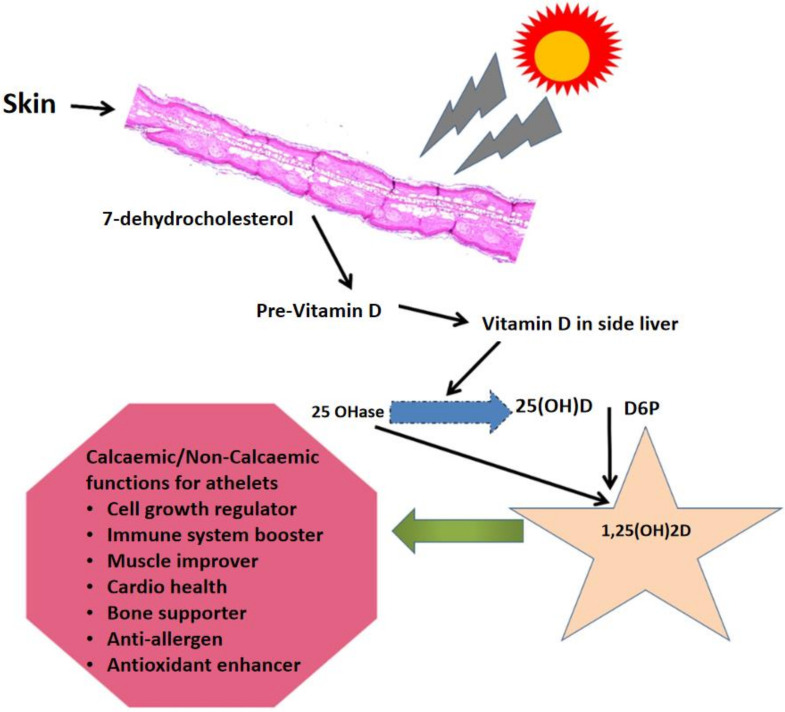
Mechanism of synthesis and functional effects of vitamin D in the bodies of human athletes and nonathletes.

**Figure 2 foods-12-00256-f002:**
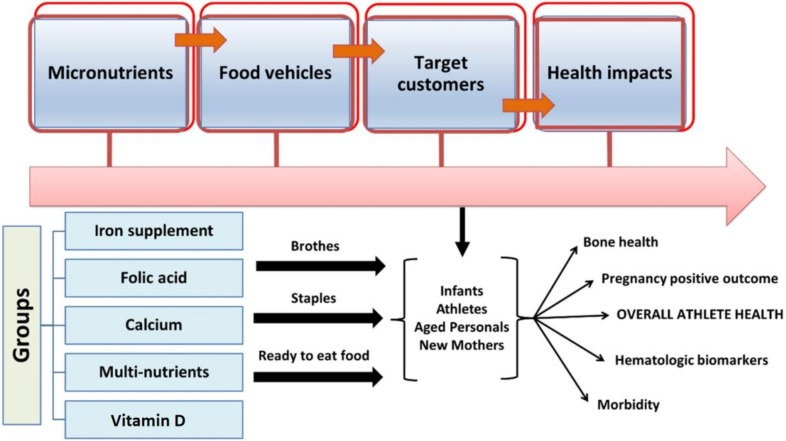
Impacts of micronutrient fortification strategies and concept-based outcomes on the health of athletes and nonathletes.

**Figure 3 foods-12-00256-f003:**
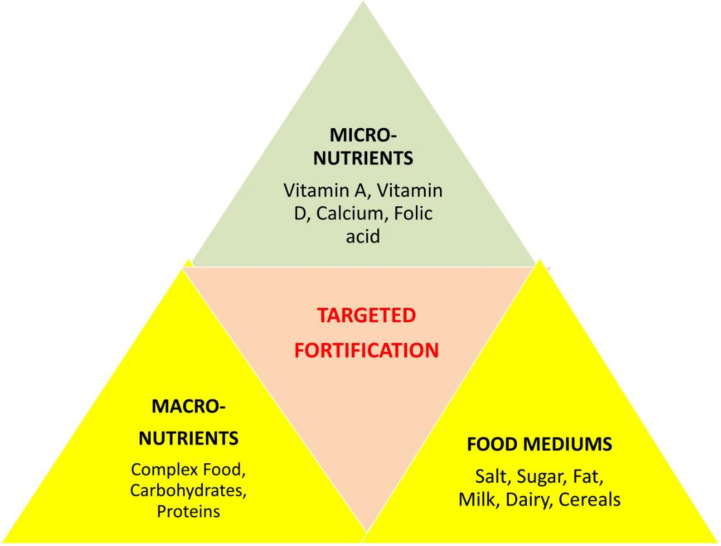
Centralized means of fortification for micronutrients, macronutrients, and food nutrients required by athletes on a daily basis.

## Data Availability

Data is contained within the article.

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
