# Peer review of "Vitamin D Fortification and Its Effect on Athletes’ Physical Improvement: A Mini Review"

_foods, 2023, doi:10.3390/foods12020256_

Round 1
Reviewer 1 Report
It would be better to adopt a methodology from PRISMA guidelines and define the review as scoping, narrative, systematic, etc..
There are chances that you did not include negative findings and PRISMA guidelines adoption and a proper review will address these issues.
What are the major limitations of this review?
What are the major implications for practice? research? clinical interventions? public health?
Author Response
Thank you very much for your high-quality review and valuable suggestion.
We apologize for the poor language of our manuscript. The language of MS have been well improved by Language Editing Services of MDPI.
All the revised portions in text are marked in yellow. The revised MS was attached in the attachment.
Please kindly check the answers we made as below:
Question 1:
What are the major limitations of this review?
Response 1:
Thanks a lot for your question.
This review has summarized recent technical advances to promote vitamin D utilization in humans during sports activities and especially in relation to athletes with daily normal practices. Data from current literatures indicates that adequate vitamin D intake contributes not only to the health of athletes but also their athletic performance. However, in this review paper, we could only provide limited information concerning the over-supplementation effect of vitamin D on human health and the athletic performance of athletes. This part will be further investigated in a future work.
Question 2:
What are the major implications for practice? research? clinical interventions? public health?
Response 2:
Thank you for your question.
The information in this review paper will have plenty of implications, such as sports medicine, public health, as well as biological chemistry and food chemistry.
It is well known that Vitamin D plays an important role in various physiological functions, and vitamin D deficiency will bring various negative effects on athletes' health and also their training efficiency. The information in this review is very helpful, which could remind sports dietician or sports physicians to regularly evaluate the vitamin D status of athletes, subsequently providing appropriate recommendations to guide athletes to achieve the best performance. In addition, the information will inspire researchers to continue to deepen the fundamental studies of vitamin D. As mentioned in the above question, over-supplementation effect of vitamin D on human health and the athletic performance was missed. Therefore, more studies are needed to determine whether vitamin D oversupplementation definitely affects the health of athletes. These studies will be helpful to explore effective ways of supplementing vitamin D-fortified foods.

Reviewer 2 Report
Vitamin D fortification and its effect on athletes’ physical improvement, a mini-review
Main observations:
The manuscript topic is consistent with the journal content.
The authors rather right concluded inter alia that “future studies should consider the efficient fortification of new products, stabilisation of vitamin D in shelf-stable products”.
LACK of LIMITATION of review-analysis (at the end of the Discussion section).
In the section “Conclusions”, there is a lack of information connected with athletes.
Lack of precision in the name of vitamin D in the manuscript body and Figure 1 and Figure 2.
Literature needs to be updated - more than 17% are articles more than ten years old - more everyday items should be used.
The major part of the discussion is consistent with the evidence and arguments and addresses the stated primary objective.
This study would be a candidate for publication in your journal as an original article with minor revisions.
Minor observations:
Lack of EXPLANATION OF ABBREVIATIONS UNDER FIGURE 1 entitled ’Mechanism of synthesis and functional effects of vitamin D in the bodies of human athletes and non-athletes” and under FIGURE 2 entitled ’ Impacts of micronutrient fortification strategies and concept-based outcomes for the health of athletes and non-athletes.’
Author Response
Thank you very much for your careful reviewing and the valuable suggestions.
The manuscript has been substantially revised according to reviewer’s suggestions. Please check the changes in the revised manuscript which have been highlighted with green color in the main text.
Please see the revised MS in attachment.
Please kindly check the answers for reviewer's question as below:
- The language and gramma havebeen well polished by Language Editing Services of MDPI.
- In the section of “Conclusions”, we have updated the discussion of vitamin D effect on athletes’physical and health, please see the changes in “Conclusions”
- Figure 1 and Figure 2 were updated according to reviewer’s suggestion.
- Literatureshave been updated. In this revised version, around 70% of the cited papers were published in recent 5 years, and 95% of the cited papers were published in recent 10 years.
The abbreviations in Figure 1 and Figure 2 have been updated with full name.

Round 2
Reviewer 1 Report
Thanks for the revisions